

# Form and function of long-range vocalizations in a Neotropical fossorial rodent: the Anillaco Tuco-Tuco (*Ctenomys* sp.)

Juan Pablo Amaya[1], Juan I. Areta[2], Veronica S. Valentinuzzi[1] and Emmanuel Zufiaurre[3]

[1] Centro Regional de Investigaciones Científicas y Transferencia Tecnológica de La Rioja (CRILAR-CONICET), Anillaco, La Rioja, Argentina
[2] Instituto de Bio y Geociencias del Noroeste Argentino (IBIGEO-CONICET), Rosario de Lerma, Salta, Argentina
[3] Grupo de Estudios sobre Biodiversidad en Agroecosistemas (GEBA), Departamento de Biodiversidad y Biología Experimental, Facultad de Ciencias Exactas y Naturales, Universidad de Buenos Aires & IEGEBA (UBA-CONICET), Ciudad Autónoma de Buenos Aires, Argentina

Corresponding author
Juan Pablo Amaya,
juanentuculandia@gmail.com

## ABSTRACT

The underground environment poses particular communication challenges for subterranean rodents. Some loud and low-pitched acoustic signals that can travel long distances are appropriate for long-range underground communication and have been suggested to be territorial signals. Long-range vocalizations (LRVs) are important in long-distance communication in *Ctenomys* tuco-tucos. We characterized the LRV of the Anillaco Tuco-Tuco (*Ctenomys* sp.) using recordings from free-living individuals and described the behavioral context in which this vocalization was produced during laboratory staged encounters between individuals of both sexes. Long-range calls of Anillaco tuco-tucos are low-frequency, broad-band, loud, and long sounds composed by the repetition of two syllable types: series (formed by notes and soft-notes) and individual notes. All vocalizations were initiated with series, but not all had individual notes. Males were heavier than females and gave significantly lower-pitched vocalizations, but acoustic features were independent of body mass in males. The pronounced variation among individuals in the arrangement and number of syllables and the existence of three types of series (dyads, triads, and tetrads), created a diverse collection of syntactic patterns in vocalizations that would provide the opportunity to encode multiple types of information. The existence of complex syntactic patterns and the description of soft-notes represent new aspects of the vocal communication of *Ctenomys*. Long-distance vocalizations by Anillaco Tuco-Tucos appear to be territorial signals used mostly in male-male interactions. First, emission of LRVs resulted in de-escalation or space-keeping in male-male and male-female encounters in laboratory experiments. Second, these vocalizations were produced most frequently (in the field and in the lab) by males in our study population. Third, males produced LRVs with greater frequency during male-male encounters compared to male-female encounters. Finally, males appear to have larger home ranges that were more spatially segregated than those of females, suggesting that males may have greater need for

long-distance signals that advertise their presence. Due to their apparent rarity, the function and acoustic features of LRV in female tuco-tucos remain inadequately known.

# INTRODUCTION

The subterranean environment has dramatic influences in the sensory biology of subterranean and fossorial rodents by imposing serious difficulties to the transmission of most communication signals (*Schleich et al., 2007*). Acoustic signals can travel long distances, depending on the frequency range and loudness of the call, and are a prime mode of long-distance communication in animals that live underground (*Heth, Frankenberg & Nevo, 1986*; *Lange et al., 2007*; *Schleich & Antenucci, 2009*). Sounds used in long-distance communication typically encode distinctive species-level information because the signal sender and the receiver cannot use other communication modes at the moment of sound emission to convey information on species identity (*Marler, 1967*). Tuco-tucos (*Ctenomys* spp.) are fossorial rodents with over 50 species found in the southern half of South America (*Lessa & Cook, 1998*). All species are adapted to a hypogeous life and exhibit parallel adaptations to other fossorial and subterranean rodents. *Ctenomys* come out to the surface to clean their burrow systems, forage in the vicinity of burrows and to disperse (*Vassallo, Kittlein & Busch, 1994*; *Tomotani et al., 2012*). These behaviours have probably facilitated the retention of fully functional eyes (*Schleich et al., 2010*), and sight and hearing are used in combination to prevent predation (*Borghi, Giannoni & Roig, 2002*; *Reig et al., 1990*).

Like other fossorial mammals, *Ctenomys* construct permanent, often elaborate tunnel systems for shelter and for securing access to food resources. The energetic cost of burrowing is considerable (*Vleck, 1979*; *Vleck, 1981*) and therefore a network of tunnels represents a valuable resource that is worth defending. Indeed, many species are extremely aggressive in defense of their burrows and intraspecific interactions among the many supposedly "solitary" underground mammals may be more common than generally supposed (*Lacey, 2000*). Most *Ctenomys* species are solitary and therefore they have to find or attract and recognize an adequate sexual partner and delimit their territories. This must be achieved without employing the typical far-reaching signals and senses normally used for these purposes by surface dwellers (*Credner, Burda & Ludescher, 1997*). While chemical and tactile channels of communication in *Ctenomys* are generally limited to use within a burrow (*Francescoli, 2000*; *Zenuto & Fanjul, 2002*), acoustic communication appears to play an important role both between individuals within the same burrow system (short-range vocalizations) and between animals in different burrow systems (long-range vocalizations (LRVs)) (*Francescoli, 1999*; *Schleich & Busch, 2002*).

Despite their ubiquity and behavioral importance, vocalizations of *Ctenomys* have been studied in some detail for only two species (*C. talarum* by *Schleich & Busch, 2002* and *C. pearsoni* by *Francescoli, 1999*; *Francescoli, 2002*). In both species, a variety of acoustic

signals are emitted for territory defense and mate attraction. LRVs in *Ctenomys* ("S-type" of *Francescoli, 1999*, and "tuc-tuc" of *Schleich & Busch, 2002*) are loud, multi-noted, and low-pitched sounds. Frequencies carrying most energy are apparently highly conserved in LRVs in this genus, suggesting that important information is encoded in the rhythmic pattern (*Francescoli, 2000*; *Francescoli, 2011*). Although LRVs are thought to be used to signal the presence of the territory owner and to assist in territory defense, the exact function of this signal has never been examined in detail.

This paper aims to: 1) provide the first quantitative acoustic characterization of the LRV of the Anillaco Tuco-Tuco (*Ctenomys* sp.) based on recordings made from free-living males and females from a population in La Rioja province, Argentina, and; 2) describe the behavioral context in which this vocalization is produced during laboratory experiments designed to assess the communicative function of this signal. In addition to contributing to our knowledge of the nature of acoustic communication in *Ctenomys*, these analyses generate support for the hypothesis that long-range acoustic signals in solitary species of tuco-tucos function in territory defense.

## MATERIALS AND METHODS

### Study site and taxon

Field studies were conducted at Anillaco (28°48′50″S, 66°55′54″O; 1,365 m asl), La Rioja, Argentina. The climate at this locality is arid with mean annual rainfall ranging from 100 to 200 mm and limited almost exclusively to the summer months (December–February) (*Abraham et al., 2009*). The soil is sandy and largely lacking organic matter, and the predominant vegetation is a shrubby steppe with characteristic Monte Desert flora dominated by species of Zygophyllaceae, Fabaceae and Cactaceae (*Abraham et al., 2009*; *Fracchia et al., 2011*).

Taxonomy in the genus *Ctenomys* is still cursory. Numerous collected specimens, field studies and audio-recordings demonstrate that our study area is occupied by a single still unidentified *Ctenomys* species informally known as the Anillaco Tuco-Tuco. Although this species has been frequently reported as *Ctenomys* cf. *knighti* (*Valentinuzzi et al., 2009*; *Fracchia et al., 2011*) or *C.* aff. *knighti* (*Tomotani et al., 2012*; *Tachinardi et al., 2014*), a formal taxonomy is lacking, and it seems likely that the Anillaco Tuco-Tuco is most closely allied to other *Ctenomys* species described from similar dry habitats in neighboring provinces (T. Sanchez, 2016, personal communication).

### Field recordings of vocalizations

We recorded vocalizations of free-living individuals of the Anillaco Tuco-Tuco during the breeding seasons (June–September) of 2014 and 2015. The breeding season was identified on the base of captures of juveniles and reproductive females (pregnant and lactating) during five consecutive years (V. Valentinuzzi, 2016, unpublished data). Recordings were made from outside the burrows in which animals vocalized using a Sennheiser ME-67 directional microphone mounted on a Rycote WS4 blimp windscreen and attached to a Marantz Professional PMD-661/MKII digital recorder. Sounds were recorded at a sample rate of 44.1 kHz and at 24 bit depth. The gain setting of the recorder

was the same for all recordings. During recording sessions, J. Amaya positioned himself among a group of inhabited burrows and waited for Anillaco tuco-tucos to vocalize. When a vocalization was perceived he walked slowly and silently approaching to the sound source. Since LRVs are unpredictable and occur underground, none of the vocalizations recorded was complete (i.e., we always missed some of the first portions of vocalizations). However, we took notes on how many series were missed, in order to characterize the composition of each bout (Fig. S1), while loosing the total duration of the vocalization. All the vocalizations were recorded from aboveground with the individuals vocalizing inside their burrow systems.

To provide critical information regarding the sex and sexual maturation of animals whose vocalizations were recorded in the field, we captured 17 of the recorded wild individuals (14 adult males and three adult females, which conformed our dataset) using plastic tube traps set at the burrow entrance closest to where each animal had been recorded. Trapped animals were transported to the laboratory where each individual was weighed, sexed, and marked with a coded michrochip (Allflex transponder, France) inserted beneath the nape skin. To ascertain that the tuco-tuco trapped was indeed the one that was recorded we proceeded as follows. The mound where the animal was captured was left open, and each animal was kept for 24 h in the laboratory. If the capture mound was still open the next day at the time of release, we were confident that the individual recorded was the same one that was captured. If the mound was closed, then at least a second individual was inhabiting the burrow system, and no certainty was possible regarding the correspondence between the trapped individual and the one recorded; in this case, the vocalizations recorded at that burrow were discarded. In total, we captured and recorded 20 individuals but discarded recordings from three that could not be identified as producing the sounds recorded as follows: first, a male and then two females were captured in the same mound; second, a female and then a male were trapped in the same mound, and third, we captured a male but could not capture the other individual in the same burrow system.

## Acoustic characterization

All acoustic measurements were made with Raven Pro 1.4 (http://www.birds.cornell.edu/raven) using the following spectrogram parameters (Window–Type: Hann, Size: 512 samples (= 10.7 ms), 3 dB Filter Bandwith: 135 Hz; Time grid–Overlap: 50%, Hop size: 256 samples (= 5.33 ms); Frequency grid–DFT size: 4,096 samples, Grid spacing: 11.7 Hz). All recordings were band-pass filtered between 80–5,000 Hz in Raven Pro 1.4 to eliminate sources of disturbance and distortion in central-tendency acoustic measurements. We manually delimited selection borders along the time axis in the waveform at points in which the waveform reached minimum energy values and changed its general repetitive pattern; this allowed us to delimit three minimum sound types in the vocalization (series note, soft-note, and individual note; Fig. 1). Using these selections we measured duration 90% of four time-delimited segments (series note duration, soft-note duration, series duration, and individual note duration), two silences between the 90% duration of sounds (silence between series and silence between individual notes), and peak frequency,

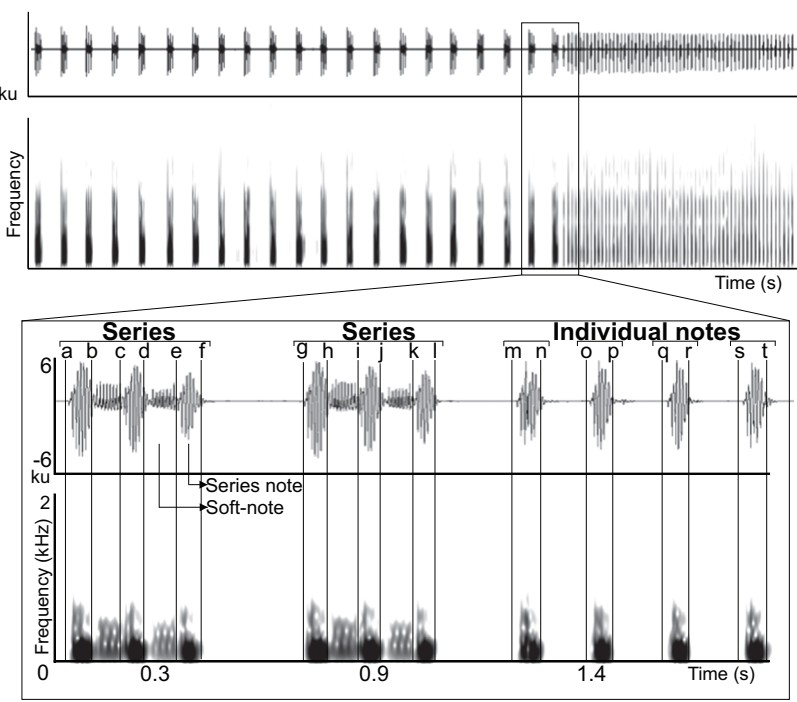

**Figure 1 Manual delimitation of syllables and silences between syllables.** Waveform and spectrogram showing manual delimitation of syllables and silences between syllables for acoustic analyses of the LRV of the Anillaco Tuco-Tuco (*Ctenomys* sp.): series note duration (a–b, c–d, e–f, g–h, i–j, k–l); soft-note duration (b–c, d–e, h–i, j–k); series duration (a–f, g–l); silence between series (f–g); individual notes duration (m–n, o–p, q–r, s–t); silence between individual notes (n–o, p–q, r–s).

IQR bandwitdh (3rd quartile frequency–1st quartile frequency), and 90% bandwidth (frequency 95%–frequency 5%) within the previously measured duration 90% of three time-delimited segments (series notes, soft-notes and individual notes) (Fig. 1, Appendix). Temporal measurements were obtained throughout the recording, and spectral measurements were obtained from 10 triad series notes and soft-notes and 10 individual notes per individual (Fig. 2). We chose to measure notes and soft-notes in triads because it was the most common and characteristic series type. We obtained acoustic measurements from one natural vocalization per individual.

For acoustic measurements of different sounds to be comparable, it is customary to choose a certain standardization threshold. Energy values below this chosen standardization threshold are discarded, and everything above it becomes the signal of interest. This would ideally result in the inclusion of the same amount of energy below and relative to the peak frequency (for frequency measurements) or below the peak amplitude of the waveform envelope (for time measurements) independently of amplitude differences in recordings of identical sounds that were recorded at different distances. However, since many of our recordings had a reduced signal-to-noise ratio we made our measurements comparable by obtaining peak frequency values and time and frequency central-tendency values for each selection, at the expense of loosing information contents. Peak frequency values are not influenced by the selection of the standardization threshold because by definition they are used to set the threshold.

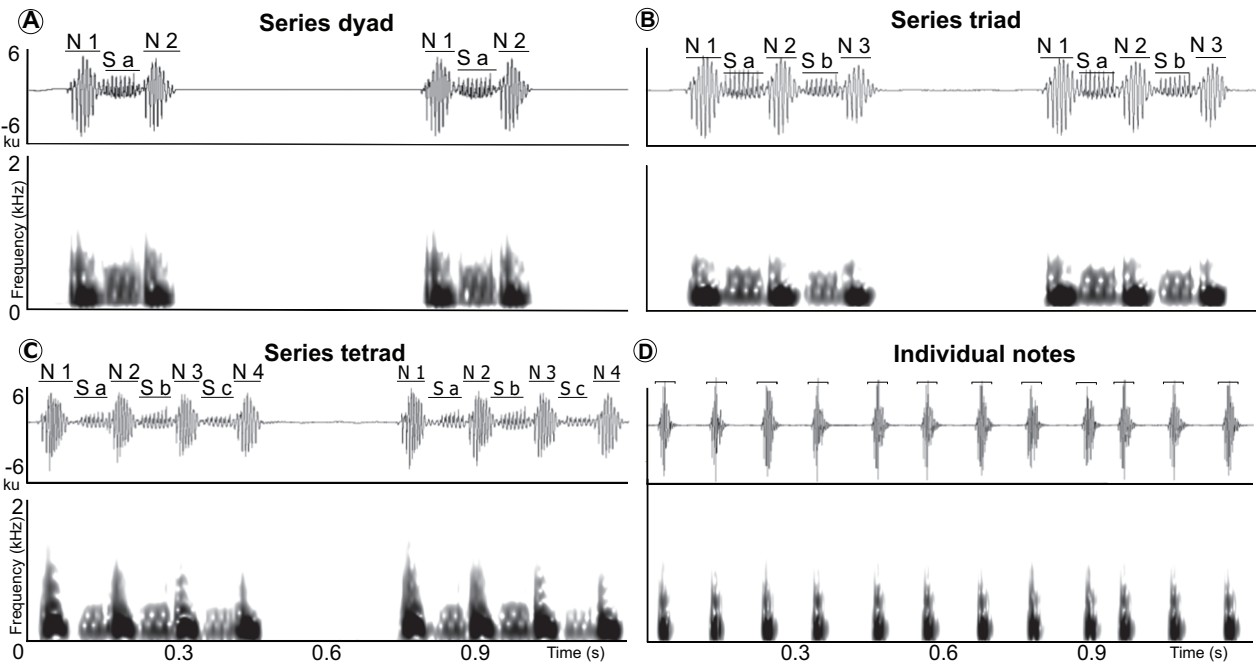

**Figure 2** **Waveform and spectrogram of the syllable types.** Waveform and spectrogram of the syllable types (three series types and individual notes) in the LRVs of the Anillaco Tuco-Tuco (*Ctenomys* sp.). (A) series dyad, (B) series triad, (C) series tetrad, and (D) individual notes. Series note (N), and series soft-note (S).

Central-tendency measurements are robust to differences in relative amplitude between sounds being compared and are based on internal cumulative-energy thresholds; results should be especially robust if energy is concentrated in a comparatively narrow band in which frequencies are similarly attenuated by the medium through which the sounds travel before being recorded. By using peak values and central-tendency measures, we circumvent the problem of making our measurements comparable in recordings with poor signal-to-noise ratio. We recognize that this implies losing the ability to obtain some comparable measures (e.g., minimum and maximum frequency), but allows us to adequately characterize the signal of interest.

## Vocalizations and body weight

To determine if production of vocal signals was influenced by body size in males, we used Pearson correlation tests (N = 14) to assess the relationship between the mean peak frequency of notes in triad series and individual notes and the weight of the male producing the call. Females were not included together with males in these analyses because of their significantly lower weight and mean peak frequencies, which might artificially result in a high correlation between weight and peak frequency by conflating the effects of sex and weight. Due to low sample size (n = 3) it was not possible to perform Pearson correlation tests separately in females.

## Laboratory studies of vocalization contexts

To learn more regarding the behavioral contexts in which LRVs are produced we recorded vocal output during experimentally manipulated encounters. Five adult male

and seven adult female Anillaco tuco-tucos were captured using plastic live traps as described above; after capture, the animals were transported to the Chronology Laboratory at CRILAR, where they were housed in an artificial burrow system consisting of four transparent glass enclosures with PVC tubes placed along the outer edges of each enclosure; the PVC tubes had been cut in half lengthwise to allow observations of the animals from outside of the experimental apparatus (Fig. S2). Each enclosure had abundant wood chips on the floor and a metallic drop door leading to the common space between enclosures. The artificial burrow system was kept in a room with a 2 m long fluorescent light source (36 W) controlled by a mechanical timer (TBCin-China) providing a light-dark cycle of 12:12 h. Light intensity during the light phase varied from 100 to 150 lux at the floor level of the enclosure. The dark phase consisted of dim light provided by incandescent red lamps (40 W) connected to a dimmer (TECLASTAR Milano, 200 W). Food was offered ad lib and consisted of sweet potatoes, carrots, sunflower seeds, oatmeal, rabbit pellets, and natural food items collected weekly from the field (*Larrea* sp., *Opuntia* sp., and *Parkinsonia praecox*). Sounds were recorded using a Zoom H4n digital hand recorder system with built-in microphones (sample rate of 44.1 kHz and 16 bit depth) and the behavior of the animals was filmed simultaneously with a digital video-camera (Nikon D5100). Video and audio recordings of behavioral interactions allowed detailed characterizations of the contexts in which vocalizations were produced.

Observations and recordings began at least one week after animals had been placed in captivity, allowing acclimatization and territory establishment in the artificial burrow system. To observe and record the associated sounds produced we staged encounters between individuals that lasted 20–40 min; all encounters were conducted during June–September in 2014, 2015 and 2016. By opening and closing the doors within the enclosure we allowed contact between animals according to experimental requirements. The experimental subjects in 2014 were two males; male A (267 g) and male B (168 g) and two females; female C (140 g) and female D (160 g), in 2015 the same two males; male A (274 g) and male B (220), and two females; female C (160 g) and female E (110 g), and in 2016 three males; male F (144 g), male G (177 g) and male H (170 g) and four females; female I (165 g), female J (137), female K (133 g) and female L (125 g). We completed 97 encounters distributed in three encounter types: 1) male-male; 2) female-female and; 3) male-female encounters (Table S1).

We characterized the behavior of individuals during the encounters using the video recordings. We analyzed behavior in 19 male-male encounters (all 15 with LRVs and four with short-range vocalizations), all five male-female and the single female-female encounters with LRVs.

We analyzed 10 min per encounter that were selected based on the intensity of interactions and on the presence of LRVs. For each encounter, we measured an instantaneous sample point every 15 s (40 sampling points per encounter) at which we assessed whether each animal was performing any of three previously defined behaviors. In decreasing level of aggression, these behaviors were: 1) Aggression: individuals fighting or at a distance less than the length of the body while displaying aggressive behavior

(individuals attacked and bit each other or individuals approached with the mouths open exposing incisive teeth); 2) Contact: individuals in contact or at a distance less than the length of the body without displaying aggressive behavior (individuals approached to sniff, mount or enter in physical contact); and 3) Independence: individuals at a distance greater than the body length (individuals ate, auto-groomed, rested, dig or moved away after an aggressive interaction). We calculated and graphed the percentage of samples per encounter in which each behavior occurred and compared how these proportions differed between the three different encounter types (male-male, male-female, and female-female).

To assess the effect of LRVs, we compared the behaviors exhibited in two 15-s sample points, one preceeding and one following the emission of vocalizations, using the previously defined behavioral categories of Aggression, Contact and Independence (Fig. S3). We quantified the number of transitions between all possible behavioral combinations to understand whether LRVs resulted in de-escalation or escalation during the encounters or whether they had no effect in altering the behavior of the individuals.

To quantify the intensity and frequency of vocal output during encounters we defined and calculated the values of three variables: probability that a staged encounter would result in a LRV (VP; number of encounters with LRVs/number of staged encounters), number of vocalizations per staged encounter (VSE; total number of LRVs/number of staged encounters) and number of vocalizations VVE; total number of LRVs/staged encounters in which vocalizations occurred).

All procedures followed the guidelines of the American Society of Mammalogists for the use of wild mammals in research (*Sikes & Gannon, 2011*). All experiments were performed at CRILAR in Anillaco and were authorized by the Environmental Department of La Rioja (permits 028–10 and 062–08) and approved by the Ethics Committee of the Faculty of Veterinary Sciences of La Plata National University, Argentina (permit 29-2-12). For statistical analyses we used InfoStat (*Di Rienzo et al., 2012*).

## RESULTS

### Description of the long-range vocalization

Long-range calls of Anillaco tuco-tucos were low-frequency, broad-band, loud, and long sounds that were composed of two syllable types: series and individual notes (Figs. 1 and 2; Table 1). The two syllable types were repeated a variable number of times during each vocalization, creating many different vocalization patterns in males and a single syntactic pattern in females (Figs. 3 and S1). The only general syntactic pattern identified was that series always preceded individual notes (Figs. 3 and S1). All vocalizations (n = 14 males and three females) had series, and only three male vocalizations did not have the individual notes (Fig. S1). For both sexes, silences between series were considerably longer than silences between individual notes (males: 1,150 ± 468 ms, n = 598 versus 180 ± 45 ms, n = 970; females: 1,075 ± 222 ms, n = 59 versus 214 ± 28 ms, n = 221).

Series had two sound types: a) notes, which were the louder and lower-pitched sounds that initiated and finished a series, and b) soft-notes, which were the softer and higher-pitched sounds that were always sandwiched by notes (Fig. 2). Depending on
**Table 1 Acoustic measurements of series and individual notes.** Acoustic measurements of triad notes and soft-notes, and individual notes from LRV in both sexes of the Anillaco Tuco-Tuco (*Ctenomys* sp.).

| | Series (Triad) | | | | | | | | | | | Individual notes | |
| | Note 1 | | Soft-note a | | Note 2 | | Soft-note b | | Note 3 | | | | | |
| Parameter | Male | Female | Male | Female | Male | Female | Male | Female | Male | Female | | | Male | Female |
|---|---|---|---|---|---|---|---|---|---|---|---|---|---|---|
| Peak Freq. (Hz) | 185.2 ± 25.7 (128.9–246.1) | 271.8 ± 10.8 (246.1–293) | 222 ± 64.1 (105.5–480.5) | 204.6 ± 42.6 (128.9–293) | 181.4 ± 22.9 (128.9–234.4) | 254.7 ± 12.3 (222.7–281.2) | 227.6 ± 57.6 (117.2–363.3) | 212 ± 33.5 (164.1–293) | 170 ± 23.6 (128.9–257.8) | 223 ± 15.8 (187.5–246.1) | | | 168 ± 20.16 (128.9–222.7) | 257.4 ± 12.5 (234.4–281.2) |
| IQR Bandwidth (Hz) | 90.7 ± 16.8 (70.3–164.1) | 109.3 ± 20.7 (93.7–175.8) | 149.0 ± 40.0 (82.1–257.8) | 152.7 ± 33.0 (128.9–175.8) | 93.1 ± 20.02 (70.3–164.1) | 114.0 ± 42.5 (82–257.8) | 142.5 ± 32.9 (93.7–222.6) | 195.6 ± 195.2 (128.9–1,218.7) | 94.1 ± 19.9 (46.9–152.4) | 100.7 ± 27.2 (82–222.7) | | | 95.7 ± 24.1 (70.3–175.8) | 101.5 ± 15.5 (82–140.6) |
| Bandwith 90% (Hz) | 263.8 ± 65.5 (187.5–398.4) | 389.8 ± 39.6 (316.4–445.3) | 399.1 ± 212.5 (234.4–2,847.7) | 412.1 ± 22.3 (351.6–457) | 254.0 ± 56.7 (187.5–398.4) | 346.0 ± 57.8 (222.7–445.3) | 394.4 ± 295.5 (257.8–383.2) | 565.2 ± 616.3 (339.8–3,433.6) | 256.1 ± 56.7 (140.6–421.9) | 318.3 ± 46.1 (199.2–375) | | | 264.6 ± 60.6 (187.5–433.6) | 346.8 ± 79.8 (210.9–445.3) |
| 1st Quartile Freq. (Hz) | 148.4 ± 25.0 (93.8–199.2) | 208.2 ± 25.1 (140.6–246.1) | 165.7 ± 42.2 (70.3–246.1) | 145.0 ± 16.9 (117.2–187.5) | 138.1 ± 21.3 (93.8–175.8) | 195.4 ± 15.3 (140.6–222.7) | 165.2 ± 42.7 (58.6–246.1) | 151.9 ± 22.9 (117.2–222.7) | 125.9 ± 18.2 (82–164.1) | 174.5 ± 14.3 (140.6–187.5) | | | 131.0 ± 18.9 (93.8–164.1) | 201.5 ± 9.9 (187.5–222.7) |
| 3rd Quartile Freq. (Hz) | 239.2 ± 31.6 (175.8–316.4) | 317.5 ± 28.0 (293–410.2) | 314.8 ± 54.3 (175.8–468.8) | 303.0 ± 25.5 (246.1–339.8) | 231.2 ± 30.6 (164.1–316.4) | 296.4 ± 24.4 (257.8–363.3) | 307.7 ± 44.2 (199.2–410.2) | 347.6 ± 210.9 (246.1–1,441.4) | 220.1 ± 31.5 (128.9–304.7) | 263.6 ± 27.9 (152.3–304.7) | | | 226.8 ± 35.7 (175.8–328.1) | 303.1 ± 24.1 (269.5–363.3) |
| Frequency 5% (Hz) | 92.2 ± 22.8 (46.9–140.6) | 157.0 ± 8.5 (140.6–175.8) | 88.9 ± 29.5 (23.4–164.1) | 57.4 ± 24.8 (11.7–93.8) | 83.5 ± 18.8 (46.9–117.2) | 138.6 ± 7.5 (128.9–152.3) | 89.8 ± 34.1 (11.7–164.1) | 53.1 ± 22.2 (23.4–93.8) | 70.3 ± 16.2 (11.7–105.5) | 116.0 ± 16.9 (58.6–140.6) | | | 71.7 ± 16.4 (35.2–105.5) | 107.0 ± 52.7 (11.7–152.3) |
| Frequency 95% (Hz) | 360.0 ± 72.3 (234.4–527.3) | 500.4 ± 65.9 (375–609.4) | 467.6 ± 43.9 (328.1–585.9) | 469.5 ± 39.4 (375–550.8) | 342.6 ± 63.4 (234.4–492.2) | 452.7 ± 70.9 (351.6–574.2) | 454.2 ± 47.6 (351.6–808.6) | 517.9 ± 253.8 (351.6–1,804.7) | 333.0 ± 58.5 (234.4–492.2) | 412.5 ± 50.9 (304.7–503.9) | | | 335.9 ± 63.1 (246.1–492.2) | 475.0 ± 91.0 (328.1–585.9) |
| Duration 90% (ms) | 33 ± 5 (26–46) | 36 ± 5 (27–48) | 61 ± 19 (32–128) | 74 ± 15 (53–112) | 32 ± 5 (21–58) | 36 ± 6 (26–53) | 61 ± 18 (37–118) | 66 ± 12 (48–96) | 33 ± 5 (21–48) | 33 ± 7 (21–53) | | | 33 ± 5 (21–48) | 38 ± 4 (32–48) |

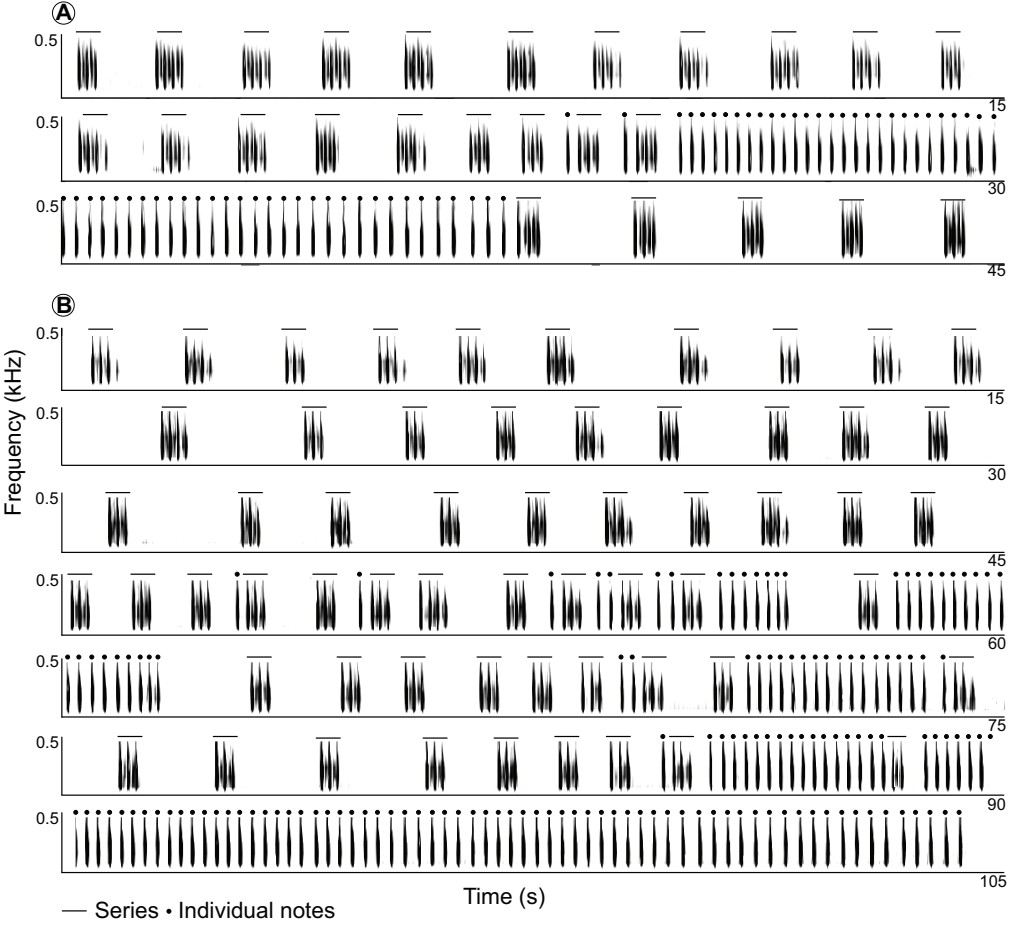

**Figure 3 Spectrograms of two full recordings of LRVs of the Anillaco Tuco-Tuco (*Ctenomys* sp.).** Spectrograms of two full recordings of LRVs of the Anillaco Tuco-Tuco (*Ctenomys* sp.) showing variation in structure and rhythm. (A) LRV of individual 9 (see Fig. S1). (B) LRV of individual 12 (see Fig. S1).

the number of notes and soft-notes, three series-patterns were identified: a) dyads, composed of two notes and one soft-note, b) triads, composed of three notes and two soft-notes, and c) tetrads, composed of four notes and three soft-notes (Fig. 2). Triads were by far the most common series-pattern for both sexes (males: 80.4%, n = 517; females: 88.8%, n = 56), followed by dyads (males: 14.9%, n = 96; females: 11.1%, n = 7), whilst tetrads were produced only by males (4.24%, n = 30). In contrast to series, individual notes had a single sound type (i.e., no soft-notes were ever documented; Fig. 2).

## Vocalizations in relation to body weight

Males were markedly heavier than females (241 ± 22.2 g vs. 143.3 ± 9.6 g, Fig. S1). All notes in triad series (the most common series type produced) and all individual notes had significantly lower mean peak frequencies in males than in females (Table 1; non-overlapping mean ± standard deviation). However, mean peak frequencies of soft-notes were indistinguishable between sexes (Table 1; extensively overlapping mean ± standard deviation). Neither the mean peak frequencies of the notes

within triad series nor mean peak frequencies of individual notes were correlated with the body weights of males (Pearson correlation tests: r = 0.08, p = 0.75 for note 1, r = −0.05, p = 0.88 for note 2, r = −0.43, p = 0.12 for note 3, and r = −0.02, p = 0.95 for individual notes).

## Behavioral context and production of long-range vocalization in captivity

LRVs were recorded 43 times in 21 of the 97 encounters staged between captive animals. All vocalizations were emitted by males, except for a single female that emitted this vocalization once (Figs. 4 and S3). The probability that a staged encounter would result in a LRV differed depending on the type of encounter (Fig. 4). The vast majority of male-male encounters (68.2% of 22) resulted in the production of LRVs which were given 36 times; in contrast, only 9.2% of 54 male-female encounters resulted in 6 such vocalizations and just 4.7% of 21 female-female encounters resulted in a single LRV (Figs. 4 and S3; Table 2). Number of VSE and VVE were higher when encounters included two males, than between a male and a female and between females (Fig. 4).

The behavior of individuals during the encounters with LRVs depended on the type of encounter (Fig. 4). Males tended to attack or more frequently to stay away from other males: male-male encounters exhibited significantly more aggression and independence, and significantly less contact behavior than other encounter types (Figs. 4 and S3). Males and females tended to stay together without conflict: male-female encounters resulted in significantly more time in contact than in independence behaviors, and exhibited a minimum amount of aggression (Figs. 4 and S3). The single female-female encounter showed a pattern similar to that of male-female encounters (Figs. 4 and S3).

LRVs resulted in de-escalating 60.5% of the times, 37% in neutral behavior and 2.5% in escalating (Table 2). The most frequent behavior exhibited after a LRV was independence, which occurred in 88.4% (38/43) of the vocal encounters, dominated by 22 aggression-independence and 15 independence-independence transitions (Table 2). Male-male vocal encounters were dominated by de-escalating (55%) and neutral (41.5%) transitions, while male-female vocal encounters were dominated by de-escalating transitions (83%) (Table 2).

Vocal encounters between males and females were characterized by a soft vocalization given by males that was not recorded from free-living animals. This "courtship" vocalization always preceded the production of a single LRV by the male, and appeared to be restricted to short-range communication between individuals. Interestingly, in the first encounters between males (A-B, F-G, G-H and F-H) they attempted to court each other, briefly giving courtship vocalizations. However, this lasted for a few moments and was followed by aggressive interactions. No courtship vocalization was observed during subsequent encounters. The fact that males briefly courted other males (as if the latter were females) during their first encounters, suggests that sexing between individuals requires from a close approach. The first male-male encounters seemed to result in sex recognition, indicating that subsequent encounters of non-naive animals were truly representative of male-male interactions. Thus, what might appear at first to
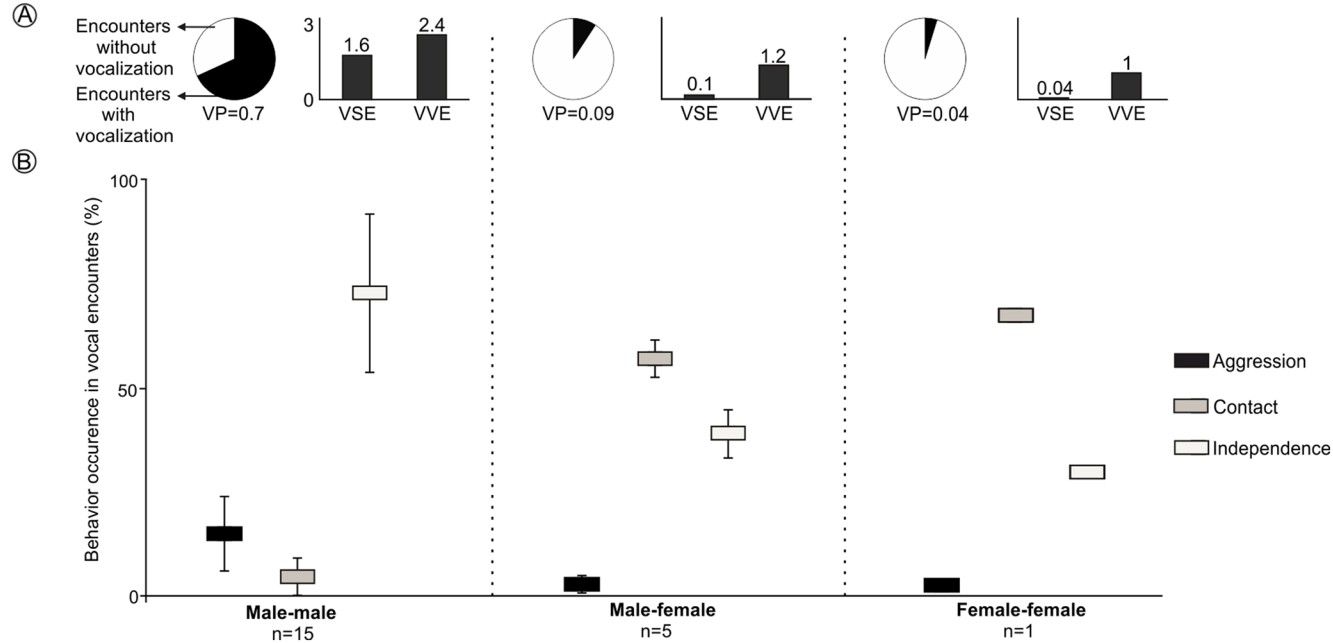

**Figure 4 Behavior of individuals of the Anillaco Tuco-Tuco (*Ctenomys* sp.) during staged encounters in captivity.** Behavior of individuals of the Anillaco Tuco-Tuco (*Ctenomys* sp.) during male-male, male-female, and female-female staged encounters in captivity. (A) Cake graph depicts the proportion of encounters with LRVs and indicates the probability of vocalization per encounter (VP), and black bars depict the number of vocalizations per staged encounter (VSE) and number of vocalizations per vocal encounter (VVE). (B) Box-plot shows mean ± SD percentage of occurrence of each behavior (Aggression, Contact and Independence) per encounter type.

**Table 2 Transitions between three behaviors (Aggression, Contact and Independence).** Number of transitions between three behaviors (Aggression, Contact and Independence) during male-male, male-female, and female-female staged vocal encounters of the Anillaco Tuco-Tuco (*Ctenomys* sp.) in captivity.

| | Behavior before and after LRV | | | | | | | | |
| | De-escalating | | | Escalating | | | Neutral | | |
| Encounter type | A-C | A-I | C-I | C-A | I-A | I-C | A-A | C-C | I-I |
|---|---|---|---|---|---|---|---|---|---|
| Male-male | 1 | 18 | 1 | 0 | 0 | 1 | 1 | 0 | 14 |
| Male-female | 2 | 3 | 0 | 0 | 0 | 0 | 0 | 0 | 1 |
| Female-female | 0 | 1 | 0 | 0 | 0 | 0 | 0 | 0 | 0 |

represent a case of pseudoreplication, becomes an important tool that shows the need of previous contact between animals to ascertain their sex-specific behavior.

In sum, males gave LRVs in proportionally more encounters and a greater number of times per encounter when facing another male than when facing a female, while females never gave LRVs in male-female encounters and did so only once in female-female encounters. While encounters between males were typically aggressive, male-female encounters appeared to result in courtship behavior.

## DISCUSSION

In this paper we have shown that long-range calls of Anillaco tuco-tucos are low-frequency, broad-band, loud, and long sounds composed by the repetition of

two syllable types: series (formed by two to four notes and one to three soft-notes) and individual notes. All vocalizations were initiated with series, but not all had individual notes, and the arrangement and number of syllables was highly variable, without exhibiting any consistent pattern. These vocalizations were sexually dimorphic and were given mostly by males and seldom by females under natural conditions. Acoustic features were independent of body mass in males, while reduced sample size precluded this evaluation in females. Results from staged encounters of captive animals suggest that LRVs occur mostly among males and are associated primarily with agonistic contexts, but future studies using a larger number of individuals should determine the generality of this pattern.

## Vocal comparisons with other ctenomyids

Two patterns of LRVs have been recognized in *Ctenomys*: Type I and Type II (*Francescoli & Quirici, 2010*). Type I vocalizations exhibit two successive segments (Parts 1 and 2), each formed by elements repeated a generally large but variable number of times conforming a single predictable pattern, while Type II vocalizations tend to be conformed by fewer elements and to occur over a shorter period of time. Descriptions of LRVs of *C. talarum* fit the Type I definition closely (*Schleich & Busch, 2002*), while those of *C. pearsoni* agree with Type II (*Francescoli & Quirici, 2010*). Overall, the general structure of the Anillaco Tuco-Tuco resembles the Type I LRV pattern. While female vocalizations fit the Type I pattern, the unpredictable and variable syntactic patterns found in males (Fig. S1) do not fit the simple two-part definition of Type I vocalizations easily. The appearance of syntactic patterns adds a new dimension to the structural characterization of vocal patterns in *Ctenomys* that has never been explored in any species in the genus.

The three types of series (dyads, triads and tetrads) found in the Anillaco Tuco-Tuco were complex structures composed by notes and soft-notes. Soft-notes have not been reported in the literature before, and their description represent another new aspect of vocal communication in ctenomyids. However, they seem to be present in several other *Ctenomys* species. For example, the note of *C. talarum* (*sensu Schleich & Busch, 2002*) consists of a conspicuous sound resembling series notes that is followed by a softer sound that may be homologous to the soft-note identified in the Anillaco Tuco-Tuco, while the typical dyads of *C. mendocinus* include a soft-note sandwiched by two notes (J. Amaya & J. Areta, 2016, unpublished data). Soft-notes differ from series notes in that they have different spectral parameters and have markedly lower relative energy values. It is important to emphasize that soft-notes are not simply reverberations from series notes. Specifically, a) we recorded some abnormal vocalizations in which the soft-notes occurred before the first note of a series, b) we recorded one series composed only of soft-notes, and c) soft-notes are not present in individual notes, as would be expected if the former were simply reverberations of the latter. Thus, we hypothesize that soft-notes are a distinctive but overlooked component of the long-distance calls of many *Ctenomys* species.

## Complexity of subterranean vocalizations

Our analyses of the structural elements of long-distance vocalizations in Anillaco tuco-tucos revealed unexpected complexity in vocal signals of these animals. The pronounced variation among individuals with regard to the number of series, the existence of three types of series (dyads, triads, and tetrads), and the variable number of individual notes created a diverse collection of syntactic patterns in vocalizations that would seem to provide the opportunity to encode multiple types of information within these calls. Here we provide the first evidence of substantial variation in the rhythmic patterns of LRVs in a single species of *Ctenomys*, a necessary pre-requisite to support the idea that important information can be encoded in the rhythmic pattern within a species (*Francescoli, 2000*; *Francescoli, 2011*).

Vocal features such as peak frequency of series-notes and individual notes and syntactic patterns were sexually dimorphic, providing an additional dimension of variability in the long-distance vocalizations of ctenomyids. Future experimental studies designed to determine what types of information are conveyed in long-distance vocalizations and how that information is encoded will substantially improve our understanding of communication within the genus *Ctenomys*.

## Functional significance of long-distance vocalizations

Long-distance vocalizations in *Ctenomys* have generally been suggested to facilitate maintenance of individual territories due to their loudness (*Francescoli, 1999*) and their structure, in particular their low frequency and long duration, both of which are considered design features typical of mammalian territorial vocalizations (*Schleich & Busch, 2002*). Territoriality may occur for multiple reasons, notably defense of resources such as food and shelter or competition for mates (*Holzmann, Agostini & Di Bitetti, 2012*). In captive *C. talarum*, long-distance vocalizations were generally given by males and occurred in agonistic contexts, leading to the conclusion that these calls were associated with male-male competition for mates (*Zenuto, 1999*; *Schleich & Busch, 2002*).

Multiple lines of evidence suggest that long-distance vocalizations by Anillaco tuco-tucos function as territorial signals to minimize aggressive encounters, especially between males. First, emission of LRVs resulted in de-escalation or space-keeping in male-male and male-female encounters. Second, these vocalizations were produced most frequently (in the field and in the lab) by males in our study population. Third, males produced LRVs with greater frequency during male-male encounters compared to male-female encounters. Finally, males in the study population appeared to have larger home ranges that were more spatially segregated than those of females (E. Lacey et al., 2016, unpublished data), suggesting that males may have greater need for long-distance signals that advertise their presence in the habitat. Supporting this, female Anillaco Tuco-Tucos gave LRVs less often than males. Due to their apparent rarity, the function and acoustic features of LRVs in female tuco-tucos remain inadequately known.

Future comparative studies of long-distance ctenomyids vocalizations will benefit from efforts to relate the acoustic features and emision of such vocalizations to the

behavioral and ecological contexts in which they occur, and will shed light on the evolution of the complex vocal communication in this Neotropical radiation of fossorial rodents.

## APPENDIX

Eight acoustic parameters measured to characterize the LRV of the Anillaco Tuco-Tuco (*Ctenomys* sp.)

1. Peak Frequency (Hz): frequency at which the peak power occurs within the selection.
2. IQR (Inter-quartile Range) Bandwidth (Hz): the difference between the 1st and 3rd Quartile Frequencies.
3. Bandwith 90% (Hz): difference between the frequency 5 and 95% frequencies.
4. 1st Quartile Frequency (Hz): frequency that divides the selection into two intervals containing the 25 and 75% of the energy in the selection.
5. 3rd Quartile Frequency (Hz): frequency that divides the selection into two intervals containing the 75 and 25% of the energy in the selection.
6. Frequency 5% (Hz): frequency that divides the selection into two frequency intervals containing the 5 and 95% of the energy in the selection.
7. Frequency 95% (Hz): frequency that divides the selection into two frequency intervals containing the 95 and 5% of the energy in the selection.
8. Duration 90% (ms): difference between the 5% time (point in the time that divides the selections into two time intervals containing 5 and 95% of the energy in the selection) and the 95% time (point in the time that divides the selections into two time intervals containing 95 and 5% of the energy in the selection).

## ACKNOWLEDGEMENTS

We are very grateful to Eileen A. Lacey for helpful comments and discussion that enriched and improved this manuscript. Marina Rutovskaya and Ema Hrouzková critically reviewed this manuscript and suggested numerous venues for improvement. One anonymous reviewer provided useful criticism of a previous version of this manuscript. Tatiana Sanchez provided information on the taxonomy of the Anillaco Tuco-Tuco. Pablo Lopez, Segundo Nuñez Campero, Carina Colque, Tatiana Sanchez and Johana Barros helped care and feed the tuco-tucos in the lab.

### Funding

Support was provided by a CONICET doctoral fellowship to J. Amaya, and sustained support was provided to Juan I. Areta and Veronica S. Valentinuzzi. The funders had no role in study design, data collection and analysis, decision to publish, or preparation of the manuscript.

## Grant Disclosures

The following grant information was disclosed by the authors:

CONICET doctoral fellowship.

## Competing Interests

The authors declare that they have no competing interests.

## Author Contributions

- Juan Pablo Amaya conceived and designed the experiments, performed the experiments, analyzed the data, contributed reagents/materials/analysis tools, wrote the paper, prepared figures and/or tables.
- Juan I. Areta conceived and designed the experiments, analyzed the data, contributed reagents/materials/analysis tools, wrote the paper, prepared figures and/or tables, reviewed drafts of the paper.
- Veronica S. Valentinuzzi contributed reagents/materials/analysis tools, reviewed drafts of the paper.
- Emmanuel Zufiaurre analyzed the data.

## Animal Ethics

The following information was supplied relating to ethical approvals (i.e., approving body and any reference numbers):

All were authorized by the Environmental Department of La Rioja (permits 028–10 and 062–08) and approved by the Ethics Committee of the Faculty of Veterinary Sciences of La Plata National University, Argentina (permit 29-2-12).

## Data Deposition

The raw data has been supplied as Supplemental Dataset Files.

## Supplemental Information

Supplemental information for this article can be found online at http://dx.doi.org/10.7717/peerj.2559#supplemental-information.

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
