# Peer review of "Form and function of long-range vocalizations in a Neotropical fossorial rodent: the Anillaco Tuco-Tuco (Ctenomys sp.)"

_PeerJ, doi:10.7717/peerj.2559_

## Round 0.1 · original submission · Major Revisions

I ask that you revise the paper according to the comments of the two reviewers and then resubmit the revised paper to PeerJ. I have marked the editorial decision "major revisions," not because there are large, substantive changes to be made, but because there are many small changes to be made, all of which add up to a pretty big total.

Reviewer 2's comments seem straightforward for you to deal with.
Reviewer 1's comments are quite extensive; my general impression is that they (the comments) point to a need for you to clarify many details in your methods and your reporting of results.

For example, reviewer 1 says says Fig. 4 does not give encounters in chronological order; is this true? She also points out that this figure does not indicate "who's doing what"---i.e., in the case of more than one animal, which vocalizations were given by which animal? Would it be possible to do this? Reviewer 1 also wants explicit descriptions of P, VSE, and VVE. Could you do a worked example for each? Does each square in this figure represent an encounter? Do the numbers in the squares = the total number of vocalizations given by both animals? What is an "n-o" encounter? I did not see that defined in the Methods section.

The comments/questions in the foregoing paragraph deal only with Fig. 4. I refrain from taking up other points raised by reviewer 1.

I look forward to seeing your revised manuscript.

·

Basic reporting

The article is written in English using clear and unambiguous text and conform to professional standards of courtesy and expression.
The article include sufficient introduction. Relevant prior literature is appropriately referenced.
But I have some comments: References furnished sloppy : missing comma in the references in the lines -96,102,104,339,342,380,382,384,386. The two authors are not mentioned in the text (line 444, 466).Line 479 (reference) - not correctly spotted padding.
The structure of the submitted article conforms to an acceptable format of ‘standard sections’ for suggested format.
Figures are relevant to the content of the article, of sufficient resolution, and appropriately described and labeled.
My comments to fig S1: in my view it is unfortunate and uninformative. Strips, indicating the number of series and individual notes virtually indistinguishable, the information on the composition of bouts given in the form of digits to the right. It's enough
My comments to fig.4.: squares, reflecting staged encounters, better arranged in chronological order.
The submission is ‘self-contained,’ and represents an appropriate ‘unit of publication’, includes all results relevant to the hypothesis.
But the results on the encounters are shown very superficially. It is not clear what individual produces vocalization during aggressive interactions: the attacked or attacking, or both individuals. How does the duration and structure of vocalization depend on the intensity of sounds, and what are the elements of behavior were accompanied by vocalization. Do males vocalized in an aggressive context in the encounters between sexes, and when: at the beginning of interactions, or sometimes in the process of courtship.

Experimental design

The submission describes original primary research.
The submission defines the research question, which are relevant and meaningful. Data on long-range vocalization in Anillaco Tuco-Tuco (Ctenomys sp.) is new and of interest.
The investigation have been conducted rigorously and to a high technical standard.
Methods described with sufficient information to be reproducible by another investigator.
But I have a lot of remarks to the method section. In the description of the recording method is not clear, if the author records the sounds from the ground or from the mouth of the burrows, which could affect the characteristics of the recorded signals. It is also unclear, several recordings were made by each individual, or only one vocalization. It would also be interesting and useful for understanding the functional significance on long-range vocalizations to note whether there are answers to the vocalization of each individual, and if so, who was responsible, at what distance? The identification between animals was made by a catching them and later used the records of only those who have been in hole alone. How many catches, in which several animals were in the hole has been registered, what sex they are? This data is much more important than the description of the area (lines 121-126). The author also writes that the age of the animals was determined (Line 150).What is the method to determine the age and why later the author does not say anything about it.
Also, the method is not specified with what statistical methods materials were analyzed
It is not clear of the methods on laboratory studies whether the animals contained transparent glass enclosure together before encounters. There is not information about animal vocalization, while sitting in separate enclosure. The diagram (Figure 4) of experiments encounters does not represent the chronology of the experiments, so it is not clear how the degree of acquaintance influences the animal vocalizations.
I recommended to make a table or chart on type and number of encounters as it’s enumerating in text is difficult to understand.
On line 242 does not match the total number of encounters: 22 -between individuals A-B, F-G, F-H, G-H (4 + 6 + 6 + 5 = 21).
There are no explanation why the four encounters, was done with 2 males and 1 female.
The experiments were recorded by video and audio equipment. However, the characteristics of sound is not shown even their structure. The biological significance of factors (P, VSE and VVE) is not explained.
The research have been conducted in conformity with the prevailing ethical standards in the field.

Validity of the findings

The data is robust, and controlled but statistically calculated insufficiently
On line 271 authors show the total duration of the vocalizations of males and females. How are they measured if no full Bout s were recorded? Furthermore conclusion that females Bout shorter is not valid because a statistical comparison is omitted. When comparing these data no statistical significant difference is get.
Lines 272-273: what the averages are taken? It is not clear what they represent and statistically this is not confirmed.
In paragraph “Vocalizations in relation to body weight” (lines 286-295) the peak frequency of the sound of male and female is compared, the statistical comparison must be done (preferably ANOVA)
The data on which the conclusions are based is provided and available in an acceptable discipline-specific repository.
Conclusions logically proved inadequate
Conclusions that long-range vocalization has aggressive context is not logical, because the sounds recorded from nature have been emitted by solitary living animals.
Structure conforms to PeerJ standard,
Raw data supplied.

Additional comments

I think we need to seriously modify the article, so the discussion and interpretation of the results in my opinion is weak and can not withstand logical analysis. However, the materials are new and interesting. So I would recommend to publish an article after correcting deficiencies
The discussing about the structure of the signal (lines 326-335) practically repeat the results. It would be interesting to discuss the possibility of the individual recognizing by vocalization, for example, to show the differences between the signals of different individuals using discriminant analysis. Expressed by the authors assumption about the nature of the signal agonist is doubtful, since vocalization was recorded in the nature of the animals that were in burrows alone.
In discussing of the complexity of underground animal vocalizations, the author suggests that they may encode multiple information. Immediately the question arises - what is it? And it is assumed that this information is embedded in the rhythm signals. So the question arises, why the author did not describe the variability of rhythm? Figure 3 shows that the intervals between the series reduced before going into a number of individual notes, and conversely, the intervals increased between individual notes before moving back into the series. If this is the case in other animals, it can be an indicator of the level of excitation of individual :( E. F. Briefer, 2012. Vocal expression of emotions in mammals: mechanisms of production and evidence // J. Zoology, 288, 1: 1-20).
It is necessary to describe in detail the behavior of animals in encounters.
The digits given in the text (line 298 and 302) do not coincide with the data shown in Figure 4
It is not clear what individual produces vocalization during aggressive interactions: the attacked or attacking, or both individuals. How does the length and structure of the vocalization correlate with intensity of interactions and what elements of behavior are accompanied by long-range vocalization. Lack of information prevents identify vocalization as aggressive or defensive. Perhaps this vocalization is a reaction to the excitement in the presence of con-specific individuals. The lack of data on the possible duets (responses vocalizations) and frequency of its emitting by individuals living together in the same hole as compared with individuals sitting in a hole alone made difficult to explain the phenomenon as an aggressive behavior.
I recommend to discuss the functional significance of the signal as a territorial. As such, it can give to neighbors information about the signal source: gender, age, the individual (the data in the statistical approach allows us to make such assumptions). This information may allow males do not meet. A female may emit a vocalization being in different physiological and motivation state - estrus, or the presence of younger at the stage when they are beginning to move.

·

Basic reporting

I would suggest to the authors to describe the tuco-tucos as fossorial (as they forage aboveground) and not subterranean (line 81 and through the text). The authors did not include all relevant articles, e.g. I miss here article of Schleich and Zenuto (2009) about sound transmission of tuco-tuco signals. The citated articles of Heth et al. (1986) and Lange et al. (2007) deal with sound transmission exclusively in the burrows, whereas tuco-tuco´s signals must spread also aboveground (line 73-75).

Experimental design

Could you please describe what criterions were used for manual selection of selection borders? (Line 188)

Validity of the findings

The comparision of long range vocalization between males and females based on vocalization of three females is not convincing. Especially if you look into Table 1 on the ranges. The data suggest that female vocalization is higher, but there is not enough females to confirm it.

Additional comments

Vocalization of tuco-tucos is very fascinating, the authors have chosen interesting field of study. However, the focus of the article could be chosen better, authors emphasize the differences between males and females, which are very low in contrast to the time structure of the vocalization. The emphasis should be shifted towards code in rhytmic pattern of long-range vocalizations. Is the rhythm individually characteristic? Does it change with the distance between individuals? The occurence of soft notes is remarkable, can it be some sort of breathing?

Here are some more specific comments for individual lines:
Line 90 ..their burrows and intraspecific interactions.. Isn´t it ..but.. instead of ..and...?
Line 93 This must be achived without .. - this statement applies on strictly subterranean species, fossorial species can use e.g. sight when foraging aboveground.
Line 133 Which species live in neighboring provinces and could be related to Anillaco tuco-tuco?
Line 164-182 This part should be shortened.
Line 304 and through the text: Be carefull in rounding the numbers, if the female vocalized one time out of 21 trials it is not 4% it is 4.76%, it would be 5% if rounded.
Line 339 How differ Type I and Type II?
Line 341 Why you mention Ctenomys talarum in this context? Is it relative to Anillaco tuco-tuco?
Line 343 How variable is the pattern of Anillaco tuco-tuco that it do not fit in Type I easily? Could you please describe and explain more the syntaxis in tuco-tuco vocalization?
Line 365 Could you describe the pronounced variation among individuals in Anillaco tuco-tuco?
Line 369 Here we provide first evidence ... I do not see any evidence in this article supporting this idea. This idea is not tested in this article at all.
Line 372-376 Information in this paragraph are included in the first paragraph of the Discussion, there is no need to repeat them.
Line 476 Correct the line with Schleich et al.
Line 479 Delete tabulator.

---

## Round 0.2 · accepted · Accept

There seems to be one tiny problem with wording in the Abstract. One sentence reads (I give the relevant part): "...created a diverse collection of syntactic patterns in vocalizations that would the opportunity to encode multiple types of information." I suppose that you left out a word, and meant to say "....that would provide the opportunity..." If you agree with me, please make the change while in Production.